# Auxin Regulates Apical Stem Cell Regeneration and Tip Growth in the Marine Red Alga *Neopyropia yezoensis*

**DOI:** 10.3390/cells11172652

**Published:** 2022-08-26

**Authors:** Kensuke Taya, Shunzei Takeuchi, Megumu Takahashi, Ken-ichiro Hayashi, Koji Mikami

**Affiliations:** 1Graduate School of Fisheries Sciences, Hokkaido University, 3-1-1 Minato-cho, Hakodate 041-8611, Japan; 2School of Fisheries Sciences, Hokkaido University, 3-1-1 Minato-cho, Hakodate 041-8611, Japan; 3Faculty of Bio-Industry, Tokyo University of Agriculture, 196 Yasaka, Abashiri 099-2493, Japan; 4Department of Bioscience, Okayama University of Science, 1-1 Ridaicho, Kita-ku, Okayama 700-0005, Japan; 5School of Food Industrial Sciences, Miyagi University, 2-2-1 Hatatate, Taihaku-ku, Sendai 982-0215, Japan

**Keywords:** tip growth, apical stem cell, branch formation, auxin, auxin antagonist, *Neopyropia yezoensis*

## Abstract

The red alga *Neopyropia yezoensis* undergoes polarized elongation and asymmetrical cell division of the apical stem cell during tip growth in filamentous generations of its life cycle: the conchocelis and conchosporangium. Side branches are also produced via tip growth, a process involving the regeneration and asymmetrical division of the apical stem cell. Here, we demonstrate that auxin plays a crucial role in these processes by using the auxin antagonist 2-(1*H*-Indol-3-yl)-4-oxo-4-phenyl-butyric acid (PEO-IAA), which specifically blocks the activity of the auxin receptor TRANSPORT INHIBITOR RESPONSE1 (TIR1) in land plants. PEO-IAA repressed both the regeneration and polarized tip growth of the apical stem cell in single-celled conchocelis; this phenomenon was reversed by treatment with the auxin indole-3-acetic acid (IAA). In addition, tip growth of the conchosporangium was accelerated by IAA treatment but repressed by PEO-IAA treatment. These findings indicate that auxin regulates polarized tip cell growth and that an auxin receptor-like protein is present in *N. yezoensis*. The sensitivity to different 5-alkoxy-IAA analogs differs considerably between *N. yezoensis* and *Arabidopsis thaliana*. *N. yezoensis* lacks a gene encoding TIR1, indicating that its auxin receptor-like protein differs from the auxin receptor of terrestrial plants. These findings shed light on auxin-induced mechanisms and the regulation of tip growth in plants.

## 1. Introduction

Tip growth is a highly polarized mode of growth involving the establishment of a single growing point at the apex of the cell, resulting in directional elongation [1,2]. Tip growth has been observed in a variety of eukaryotic taxa including fungi, oomycetes, green and brown algae, and terrestrial plants [1,3,4,5,6]. Filamentous organisms exhibit two types of tip growth: the single-cell type involving polarized elongation of a single filamentous cell (such as in root hairs and pollen tubes of angiosperms) [7,8,9,10] and the hyphae type involving continual production of new tip cells through polarized elongation and the subsequent asymmetrical division of the apical cell (such as in protonemata and rhizoids of streptophyte algae, bryophytes (liverworts, hornworts, and mosses), and ferns) [4,11,12,13,14]. The regulatory mechanisms of tip growth have been extensively investigated using root hairs and pollen tubes as single-cell models. These mechanisms involve ion flux, the asymmetric distribution of F-actin, phosphoinositides, membrane trafficking of membrane and cell wall materials, the production of reactive oxygen species, and plant hormones [15,16,17,18,19,20,21,22,23,24,25].

The hyphae of fungi and protonemata of mosses are multicellular, uniseriate, cylindrical structures whose growth occurs only in the tip cells. Tip cell elongation is restricted to the apex, and asymmetric division of an elongated tip cell produces a new apical tip cell and a basal nondividing cell [15,26,27]. Since these events occur at the apex of the growing cells, a single growing point is established at the apex and the growth direction is polarized, a process responsible for tip growth. During hyphae-type tip growth in the protonema of the moss *Physcomitrium patens*, the tip cell functions as a stem cell, as it produces two types of cells with each division: a new apical cell and a differentiated non-dividing cell [12]. Such filamentous organisms also produce side branches by generating new tip-growing apical cells from nondividing differentiated cells; these new apical cells are then maintained as stem cells during the tip growth of the side branches [4,12]. Thus, the regeneration of apical stem cells from non-stem cells results in the formation of branched colonies.

The plant hormone auxin plays important roles in tip growth. Exogenous treatment with the auxin indole-3-acetic acid (IAA) stimulates the tip growth of moss protonema and angiosperm pollen tubes [28,29,30]. Mutants in auxin signaling exhibit repressed root hair growth. By contrast, overexpressing genes involved in the perception, transport, and metabolism of auxin leads to enhanced root hair growth [22,31,32]. In addition, IAA induces the degradation of AUXIN/INDOLE-3-ACETIC ACID (Aux/IAA) transcriptional repressors. Specifically, IAA binds to TRANSPORT INHIBITOR RESPONSE (TIR1), an auxin receptor that functions as the F-box subunit of the ubiquitin ligase complex AUXIN SIGNALING F-BOX (SCF^TIR1/AFB^), thereby releasing Aux/IAA from auxin-responsive transcription factors (ARFs) to activate IAA-inducible gene expression [33,34,35].

This TIR1-mediated auxin-induced activation of ARFs was recently shown to function in tip growth. High endogenous auxin levels in the root hair differentiation zone triggered the expression of *ARF19* under low phosphate conditions, which stimulate root hair production and elongation [36]. In addition, ARF7 and ARF19 interact with the promoter of *ERULUS* (*ERU*) and activate its expression; *ERU* encodes a plasma membrane-localized receptor-like kinase that is involved in fine-tuning cell wall composition in root hairs in *Arabidopsis thaliana* [37]. These findings point to a direct link between auxin signal transduction and tip growth.

The marine red alga *Neopyropia yezoensis* is a representative species used for nori aquaculture in Asian countries such as Japan, South Korea, and China [38]. Despite the successful establishment of industrial large-scale aquaculture systems for *N. yezoensis* at the sea surface, our biological understanding of this alga is limited, and the regulatory mechanisms of its growth, life cycle, development, and environmental stress responses are mostly unknown. We recently demonstrated that tip growth is essential for maintaining the proliferation of the filamentous generations of *N. yezoensis*, including the conchocelis (sporophyte generation) and conchosporangium (conchosporophyte generation) [39,40,41,42]. In these generations, only the apical tip cell undergoes elongation and division, which results in the production of two different cell types: a copy of the apical cell at the tip of the filament and a neighboring differentiated nondividing cell [39,40,41]. 

Like the tip cells of *P. patens* protonema [12], the tip cells of the conchocelis and conchosporangium of *N. yezoensis* are also thought to be stem cells, suggesting that these filamentous generations are ideal models to explore the mechanisms regulating tip growth and the regeneration of tip-growing apical cells in seaweeds. Therefore, elucidating the mechanisms underlying the production and maintenance of the apical stem cell in *N. yezoensis* would be beneficial for understanding tip growth as a growth strategy in marine filamentous photosynthetic organisms.

Here, we established novel experimental systems to study tip growth and investigated the role of auxin in tip growth in conchocelis and conchosporangium filaments using artificially synthesized auxin-derived compounds [33,43,44,45]. Our results demonstrate the critical role of auxin in tip growth in the filamentous generations of *N. yezoensis*.

## 2. Materials and Methods

### 2.1. Algal Materials and Culture Conditions

Two filamentous generations in the life cycle of *N. yezoensis* (strain U-51), conchocelis (sporophyte) and conchosporangium (conchosporophyte), were maintained in sterilized artificial seawater as described by [46]. The cultures were grown under 60 μmol photons m^−2^ s^−1^ light with a long-day photoperiod (14-h light/10-h dark) at 15 °C and aerated with air filtered through a 0.22-μm filter (Whatman, Maidstone, UK). The culture medium was changed weekly.

### 2.2. Synthesis of Chemical Compounds

Auxin antagonists PEO-IAA [2-(1*H*-Indol-3-yl)-4-oxo-4-phenyl-butyric acid], 4-Cl-PEO-IAA [2-(1*H*-Indol-3-yl)-4-oxo-4-(4-chlorophenyl)-butyric acid], and BH-IAA [8-(tert-Butoxycarbonylamino)-2-(1*H*-indol-3-yl)octanoic acid], and 5-alkoxy-IAAs were synthesized according to the methods described in [33,43,44,45].

### 2.3. Preparation of Single-Celled Conchocelis and Conchosporangium

To excise single-celled conchocelis and conchosporangium, these multicellular filamentous structures were chopped with a razor blade, filtered through a 10-μm nylon mesh to remove large pieces, and incubated in 9-cm dishes (Asnol dish 90 mm (diameter) × 20 mm (height), As One, Osaka, Japan) containing 30 mL seawater at 15 °C for 10 min. Cells whose adjacent cells disappeared were picked up with micropipettes under an Olympus IX73 light microscope equipped with an Olympus DP22 camera, transferred into 96-well plates (one cell/well containing 200 μL artificial seawater with or without chemicals, as indicated), and analyzed after 1 week of culture under the conditions described above but without aeration. The branching rate was calculated as the number of cells producing a branch as a percentage of total number of cells observed. 

### 2.4. Observation of Naturally Produced Conchosporangia

Single-celled conchocelis were statically cultured in wells containing sterilized artificial seawater under the conditions described above except that they were aerated. Swelling tip cells of conchocelis side branches were then detected by microscopy, and their growth and side-branch formation were monitored for 7 days using an Olympus IX73 light microscope equipped with an Olympus DP22 camera. The branching rate was calculated as the number of cells producing a branch as a percentage of the total number of cells observed.

### 2.5. Chemical Treatment of Isolated Cells and Naturally Produced Conchosporangia

Pharmacological treatments with auxins, auxin antagonists, or 5-alkoxy-IAAs were performed by incubating isolated cells and naturally produced conchosporangium filaments at 15 °C in wells containing 200 μL artificial seawater. Auxin treatment was performed by incubating the cells for 1 week in 5, 10, 30, 50, or 100 μM IAA (Nakalai Tesque, Kyoto, Japan), NAA (Sigma-Aldrich, Merck KGaA, Darmstadt, Germany), or 2,4-D (Nakalai Tesque, Kyoto, Japan) in 200 μL artificial seawater in wells. Three types of auxin antagonists, PEO-IAA, 4-Cl-PEO-IAA, and BH-IAA [33,43], were added to the artificial seawater in the wells to generate 10, 20, 30, 40, or 50 μM solutions, followed by incubation of the cultures for 1 week. For co-treatment with auxins and PEO-IAA, different concentrations of IAA, NAA, or 2,4-D (5, 10, 20, or 30 μM) and a single concentration of PEO- IAA (30 μM) were used. Five derivatives of 5-alkoxy-IAAs, named **1a** to **5a** [45], were used at a concentration of 30 μM for the assays, whereas the other experiments employed 10 μM 5-alkoxy-IAAs or IAA in the presence of 30 μM PEO-IAA for 2 weeks. Cells treated with chemicals for 2 weeks in static culture were observed and photographed using an Olympus IX73 light microscope equipped with an Olympus DP22 camera to evaluate the effects of chemicals on tip growth and the formation of side branch initials. Branching rate was calculated as the number of cells producing a branch as a percentage of the total number of cells observed.

### 2.6. Statistical Analysis

Mean values ± SD were calculated from triplicate experiments. A statistically significant interaction was detected between the duration of incubation with various combinations of chemicals and the regeneration of apical stem cells, as determined by one-way ANOVA with the Tukey–Kramer test (*p* < 0.05). Significant differences for each set of treatments were determined using a cutoff value of *p* < 0.05. 

## 3. Results

### 3.1. Generation of Apical Stem Cells from Single-Celled Conchocelis to Observe Tip Growth

Since side branches develop from differentiated nondividing cells in conchocelis filaments (see Figure 1B of [47]), we reasoned that tip growth, with the production and maintenance of an apical stem cell, could be observed by examining the formation of a side branch from a single-celled conchocelis. Thus, we prepared single-celled conchocelis (Figure 1B) by chopping sporophyte filaments (Figure 1A) with a razor blade and examined branching from these isolated cells. As expected, approximately 90% of nonbranched single cells produced side branches after culturing for 7 days (Figure 2G); the positions of side branch initials in the cylindrical conchocelis cells appeared to be random (Figure 1C–E). Therefore, the single-celled conchocelis provides a novel, simple experimental system for addressing the regulatory mechanisms of tip growth of the filamentous sporophyte generation of *N. yezoensis*.

### 3.2. The Role of Auxin in the Generation and Tip Growth of Side Branches in Conchocelis

Since the exogenous application of auxin has positive effects on tip growth in terrestrial plants [28,29,30], we first examined whether tip growth from a single-celled conchocelis could be stimulated by the exogenous application of indole-3-acetic acid (IAA). However, no effect was observed (data not shown), suggesting that if auxin is required for tip growth, an adequate amount was already present in the isolated cells. 

We next addressed the role of auxin in tip growth by examining the effect of modulating the activity of the auxin receptor, since the functions of auxin receptors can be elucidated via chemical biology approaches using auxin receptor antagonists [33,43,44]. When isolated conchocelis cells were treated with the auxin receptor antagonist PEO-IAA [33], the generation and tip growth of side branches were repressed in a concentration-dependent manner (Figure 2). The repression of branch formation by treatment with 30 μM PEO-IAA was recovered by the exogenous application of 5, 10, or 20 μM IAA (Figure 3). Although 30 μM IAA had a negative effect on branch formation (Figure 3), this was likely an off-target effect of a high concentration of IAA on growth, as observed in terrestrial plants [48,49,50,51]. These findings indicate that auxin plays an important role in regulating tip growth and that *N. yezoensis* contains an auxin receptor-like protein that regulates tip growth in conchocelis filaments.

### 3.3. Role of Auxin in Tip Growth and Side Branch Production in Conchosporangia

The conchosporangium, a structure representing the conchosporophyte generation of the *N. yezoensis* life cycle [40], is produced on the conchocelis via the swelling of the apical cell of a side branch [52], representing a type of tip growth [47]. Since nondividing differentiated cells produce side branches (Appendix A), we expected that single-celled conchosporangium (as well as conchocelis) would be suitable to study tip growth. However, most single cells did not produce side branches, unlike the single-celled conchocelis, as described above (Figure 4A). Nonetheless, isolated cells derived from apical cells were able to divide and generate side branch initials for their tip growth (Figure 4B). As shown in Figure 4B, branch formation from the conchosporangium exhibited two unique characteristics. First, the branches formed at the nondividing differentiated cell adjacent to the apical tip cell. Second, the width of the branch and the length of the nondividing cell were similar, resulting in the production of thick, cylinder-shaped outgrowths.

Chopping the conchosporangia did not produce enough free apical cells to allow us to perform experiments. Thus, we improved the experimental system using single-celled conchocelis. In these structures, the formation of conchosporangia was observed in some branches after about 2 weeks of culture. This spontaneous production of conchosporangia enabled us to monitor tip growth more easily compared to examining isolated apical cells obtained by chopping. As shown in Figure 5A,B, we were able to confirm the same pattern of tip growth and branch formation in these conchosporangia as we had observed in single-celled conchosporangia (Figure 4B). 

We employed our novel experimental system to investigate the role of auxin in the tip growth of conchosporangia. When the tips of side branches that formed on conchocelis filaments began to swell, which we took as a signal of conchosporangium development, we treated the filaments with 5, 15, or 30 μM IAA and observed growth after 3, 5, and 7 days of culture. In contrast to conchocelis, the tip growth of conchosporangium was accelerated by exogenously supplied auxin. The greatest increases in both the length and cell number of branches occurred following 15 μM IAA treatment, although 5 and 30 μM IAA treatments had lesser but significant effects equally (Figure 5C,D). By contrast, treating the swelling tips of side branches with 5, 10, or 20 μM PEO-IAA resulted in the repression of tip growth in a concentration-dependent manner (Figure 6). These findings indicate that auxin regulates tip growth in conchosporangia.

### 3.4. Characterization of a Unique Auxin Receptor in N. yezoensis Using IAA Derivatives

The auxin receptor antagonist PEO-IAA repressed the regeneration and tip growth of apical tip cells in side branches from both conchocelis and conchosporangium, suggesting the presence of functional auxin receptors in *N. yezoensis*. However, in previous studies, we failed to identify genes encoding homologs of the auxin receptor TIR1 or auxin-responsive factors (ARFs) in the *N. yezoensis* genome [53,54]. We therefore predicted that *N. yezoensis* might contain a novel, unknown auxin receptor-like protein. To test this hypothesis, we utilized chemical biology approaches. 

First, we assessed the binding of auxin to an unknown auxin receptor by examining the reversal by auxins of the repression of side branch formation by 30 μM PEO-IAA. The native auxin IAA exhibited potent auxin activity in the promotion of branch formation (Figure 7), whereas 5 to 30 μM 2,4-D or NAA showed weak activity in branch formation in the presence of PEO-IAA (Appendix A). These findings indicate that both native and synthetic auxins function in *N. yezoensis*, demonstrating the presence of an auxin receptor-like protein in this alga. 

We then examined the binding of another auxin antagonist, the PEO-IAA derivative 4-Cl-PEO-IAA [33,43], to the auxin receptor-like protein. As shown in Appendix A, 4-Cl-l-PEO-IAA acted as an auxin antagonist for side branch formation, indicating that the IAA moiety of PEO-IAA is recognized by an unknown auxin receptor-like protein in *N. yezoensis*. When the auxin receptor antagonist BH-IAA, which is a structural derivative distinct from PEO-IAA and 4-Cl-PEO-IAA [33,43], was employed, it showed weak auxin-antagonistic activity on side branch formation in *N. yezoensis* (Appendix A). These results indicate that an auxin receptor-like protein recognizes the common IAA moiety of these auxin antagonists. 

We next examined the auxin activity of 5-alkoxy-IAAs (see Figure 1 in [45]) against the *N. yezoensis* auxin receptor-like protein. In *A. thaliana*, 5-alkoxy-IAAs such as 5-methoxy-IAA (**1a**), 5-ethoxy-IAA (**2a**), 5-propoxy-IAA (**3a**), and 5-butoxy-IAA (**4a**) showed auxin activity by binding with TIR1 to induce auxin-responsive gene expression, whereas 5-pentoxy-IAA (**5a**), 5-hexyloxy-IAA (**6a**), and 5-benzoxyl-IAA (**7a**) lacked TIR1-binding activity [45]. Thus, we examined the auxin activity of **1a** to **5a** in *N. yezoensis*. When isolated conchocelis cells were incubated with 30 μM 5-alkoxy-IAAs for 2 weeks, branching and tip growth occurred normally in the presence of **1a** and **2a** but was partially repressed by **3a** and **4a** (Figure 7A). By contrast, **5a** strongly repressed branching, as did PEO-IAA (Figure 7A). 

In treating single-celled conchocelis with 30 μM PEO-IAA plus 10 μM 5-alkoxy-IAAs for 2 weeks, treatment with **1a** and **2a** allowed the recovery from the inhibited branching and tip growth induced by PEO-IAA, and by 10 μM IAA (Figure 7B). However, **3a** and **4a** did not counteract the effects of PEO-IAA, and **5a** showed additive antagonistic activity with PEO-IAA (Figure 7B). Thus, **1a** and **2a** acted as auxins, while **3a**, **4a**, and **5a** displayed auxin-antagonistic activity, indicating that **3a**, **4a**, and **5a** act differently in *N. yezoensis* than in *A. thaliana*. Notably, although **5a** apparently bound to an auxin receptor-like protein in *N. yezoensis* and functioned as an auxin antagonist, a previous study showed that this compound did not affect TIR1 function in *A. thaliana* [45]. Therefore, based on the results of our auxin structure-based experiments, we conclude that the auxin-recognition site of the auxin receptor-like protein of *N. yezoensis* is different from that of the *A. thaliana* TIR1/AFB receptor. Thus, auxin regulates side branch formation and tip growth in *N. yezoensis* via an unknown auxin receptor.

## 4. Discussion

We performed chemical biology studies to explore the mechanisms underpinning tip growth of the conchocelis and conchosporangium in the red seaweed *N. yezoensis*. Our novel experimental procedures allowed us to successfully demonstrate the critical role of auxin in tip growth of two filamentous generations in the *N. yezoensis* life cycle: conchocelis (sporophyte) and conchosporangium (conchosporophyte). Despite the presence of auxin in *N. yezoensis* and ‘*Bangia*’ sp. ESS1 [53,54], the physiological roles of auxin in Bangiales have not yet been elucidated. Thus, our results provide the first evidence for the physiological function of auxin in Bangiales. Since auxin was already shown to be involved in tip growth in the brown alga *Ectocarpus siliculosus* [5], it is plausible that role of auxin in regulating tip growth is conserved in seaweeds.

As shown in Figure 1, the positions of branches in single-celled conchocelis were different from those in the protonema of *P. patens*, whose branch initials are produced at the apical ends of filamentous differentiated cells [55]. Our findings indicate that polar regulation does not determine the branching position in isolated conchocelis cells. Since it is unknown whether the isolation of single cells affects the positioning of apical stem cell production, it is important to examine branch formation in natural filamentous conchocelis to address whether polar regulation affects branch formation in *N. yezoensis*. 

While the conchocelis and conchosporangium both expanded via tip growth and branching in *N. yezoensis*, our observations indicate that their growth processes are different, especially their branching patterns. In conchocelis filaments, thin hyphae-type outgrowths are produced in nondividing mature cells distant from the tip cell in the primary filament. Since the nondividing cell is longer than the width of cells in the primary filament, branching usually involves tubular filamentous extensions, similar to branch formation in protonema of the moss *P. patens* [4,12,56]. By contrast, branching in conchosporangia involves the production of thick outgrowths whose diameters are nearly identical to the length of the nondividing cell (Figure 4 and Figure 5), like in *E. siliculosus* [4,5]. These findings point to the different mechanisms by which branching is achieved in the conchocelis and conchosporangium: The former requires the side branch initial to be positioned somewhere along the longitudinal side of the cell, but the latter does not. Although common protein factors function in tip growth in both protonema and caulonema in *P. patens* [56], perhaps the regulatory machinery of tip growth is basically similar but not identical in the conchocelis and conchosporangium. Thus, further characterizing the factors involved in the generation of the novel apical stem cell in branching initials will help uncover the differences in the branching systems of these two generations of the *N. yezoensis* life cycle.

Our results also indicate the presence of a novel, yet to be identified, auxin receptor in *N. yezoensis*. Genes encoding the auxin receptor TIR1/AFB and the transcription factors Aux/IAAs and ARFs have not been identified in the genomes of Bangiales species [53,54]. Thus, factors that participate in the auxin signal transduction pathway in *N. yezoensis* were not previously identified and appear to be novel [53,54]. Our chemical biology studies clearly demonstrated that the auxin receptor-like protein of *N. yezoensis* binds to the **1a** to **5a** versions of 5-alkoxy-IAA used in the present study; this is markedly different from *A. thaliana* TIR1/AFB, which does not recognize **5a** [45].

Since **1a** and **2a** showed auxin activity and **5a**, like **3a** and **4a** (Figure 7), acts as an auxin antagonist, it appears that the *N. yezoensis* auxin receptor-like protein contains one auxin-binding pocket. However, this hypothesis does not explain why treating single-celled conchocelis with PEO-IAA and **5a** together had additive effects repressing side branch formation (Figure 7B). If these molecules competitively bind to the receptor-like protein with the same affinity, the effects of their combined application should be identical to those of PEO-IAA alone. Thus, the affinity of **5a** for the *N. yezoensis* auxin receptor-like protein should be higher than that of PEO-IAA. However, as shown in Figure 7A, PEO-IAA and **5a** had identical effects in terms of antagonizing side branching. It is therefore plausible that *N. yezoensis* contains at least two auxin receptor-like proteins, one of which dominantly binds to PEO-IAA and another to **5a**. In this case, the additive effect of these chemicals in the combined treatment of single-celled conchocelis could be explained by the combined actions of two auxin receptor-like proteins binding separately to PEO-IAA and **5a**. Indeed, the TIR1/AFB families of terrestrial plants contain multiple, functionally diverse auxin receptors [57,58,59]. Like *N. yezoensis* [53,54], *E. siliculosus* also lacks any factors homologous to known auxin signal transduction components in terrestrial plants [60], indicating that the modes of action of auxin in regulating tip growth in *N. yezoensis* and *E. siliculosus* are different from those in terrestrial plants. Therefore, identifying and characterizing auxin receptors and factors involved in the auxin signal transduction pathway in seaweeds would enable us to define the novel regulatory mechanisms of auxin-directed tip growth in photosynthetic filamentous organisms.

Notably, the *N. yezoensis* genome also lacks homologs of the genes encoding auxin biosynthetic enzymes in *A. thaliana* [53,54]. Thus, the origin of the auxin in this apparently non-auxin-producing seaweed must be clarified to understand how it obtains this plant hormone to regulate tip growth in both the conchocelis and conchosporangium. A variety of epiphytic bacteria have been isolated from *N. yezoensis* [61,62,63,64,65,66] and other red seaweeds [67,68,69,70], some of which synthesize IAA [67,71]. Although the physiological function of bacterial IAA has not yet been confirmed, it is possible that epiphytic bacteria of *N. yezoensis* synthesize IAA and promote the generation of apical stem cells and their expansion for tip growth in conchocelis and conchosporangium cells. Moreover, since *E. siliculosus* can produce auxin [60] but hosts many types of epiphytic bacteria [72,73], the supply of auxin from epiphytic bacteria might depend on the algal species. Therefore, the origin and modes of action of auxin in *N. yezoensis* should be addressed to elucidate the regulatory mechanisms of tip growth in seaweeds.

## 5. Conclusions

The tip growth in both conchocelis and conchosporangia filaments was repressed by auxin antagonists and sensitivity to 5-alkoxy-IAAs was different between *N. yezoensis* and *A. thaliana*. Thus, our chemical biology studies indicated that the tip growth of filamentous generations of the *N. yezoensis* life cycle is regulated by auxin via a novel unknown auxin receptor-like protein. However, even though the outgrowth of secondary filaments in both generations occurs via tip growth, their branching patterns differ. In addition, it is unknown how *N. yezoensis* obtains auxin. To clarify the molecular basis of the auxin-mediated regulation of tip growth in *N. yezoensis*, it is important to identify the auxin receptor and factors that mediate auxin signaling as well as auxin-producing epiphytic bacteria that might be involved in regulating tip growth.

## Figures and Tables

**Figure 1 cells-11-02652-f001:**
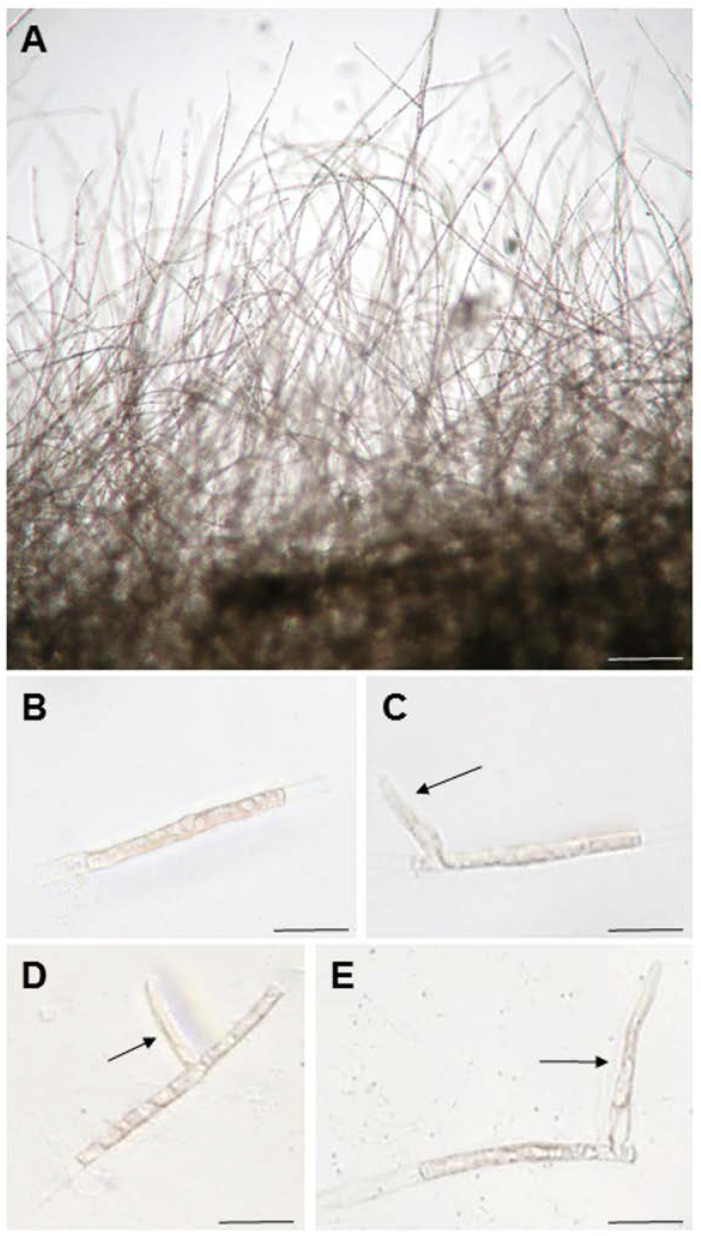
Generation of tip-growing side branches from single-celled conchocelis filaments. (**A**) Conchocelis filaments maintained in the laboratory; (**B**) a single-celled conchocelis; (**C**–**E**) side branches produced at various positions on single-celled conchocelis. Arrows indicate side branches. Scale bars: 100 μm in (**A**); 25 μm in (**B**–**E**).

**Figure 2 cells-11-02652-f002:**
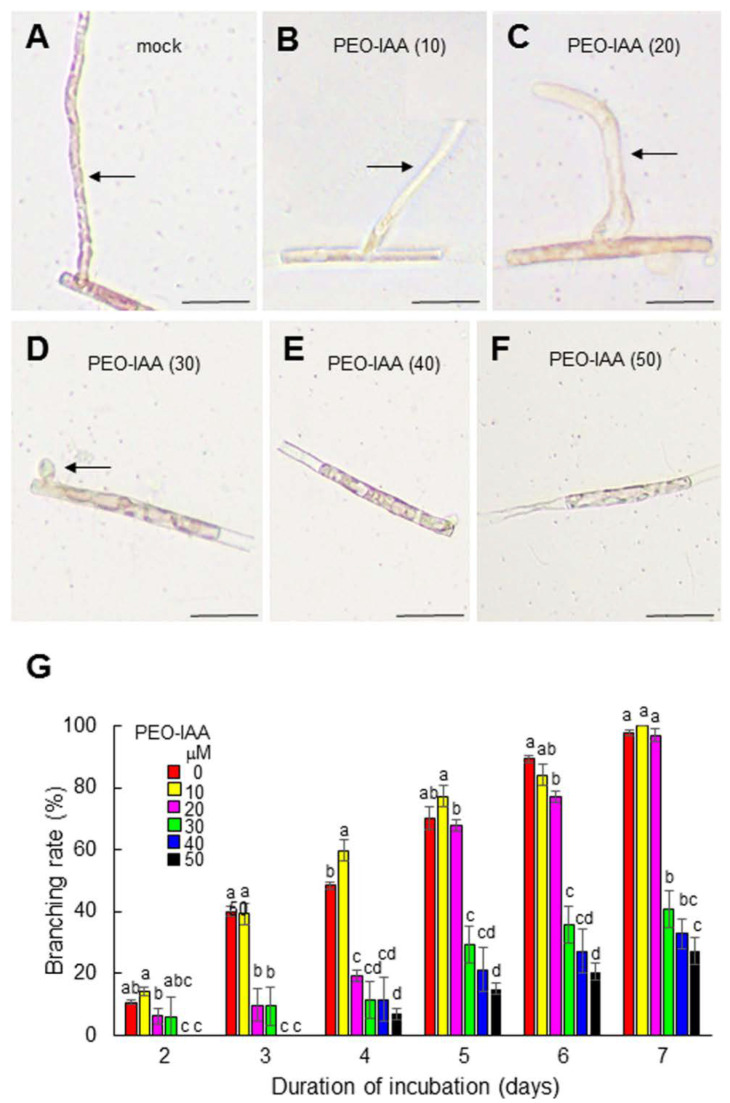
Effects of treatment with the auxin antagonist PEO-IAA on side-branch production in a single-celled conchocelis. (**A**–**F**) Photographs of typical effects of the auxin antagonist on the growth of side branches from single-celled conchocelis treated with 0, 10, 20, 30, 40, or 50 μM PEO-IAA for 7 days. Arrows indicate side branches. Scale bars: 50 μm. (**G**) Changes in branching ratios following the treatment of single-celled conchocelis with various concentrations of PEO-IAA. Side branch formation was observed in single-celled conchocelis cultured in various concentrations of PEO-IAA for 7 days, and the number of conchocelis with branches at 2, 3, 4, 5, 6, and 7 days after treatment was counted under a microscopic to calculate the branching rate. Error bars indicate the standard deviation of triplicate independent experiments (*n* = 3), and different lowercase letters denote significant differences in branching rate, as determined by Tukey’s test (*p* < 0.05) for each set of incubation times.

**Figure 3 cells-11-02652-f003:**
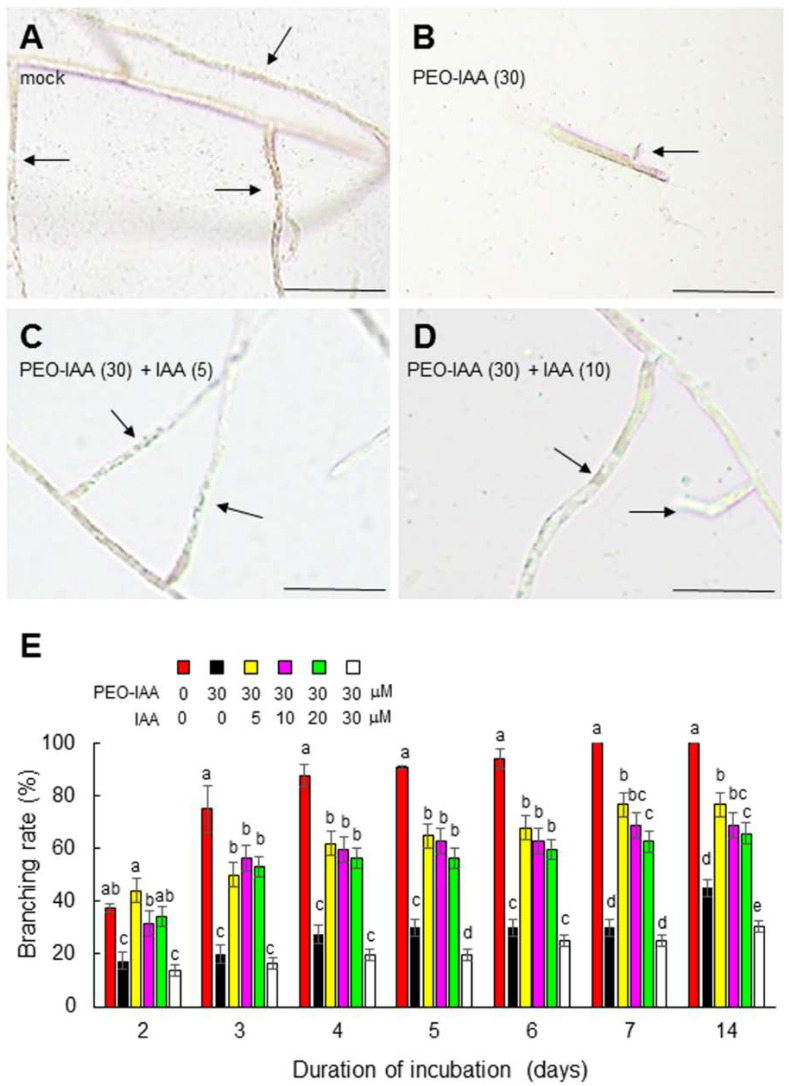
Reversal of the inhibitory effects of PEO-IAA following exogenous treatment with IAA. (**A**) Growth of side branches produced from a single-celled conchocelis following 2 weeks of culture. Arrows indicate side branches. Scale bars: 50 μm. (**B**–**D**) Photographs of typical single-celled conchocelis treated with 0, 5, or 10 μM IAA for 7 days in the presence of 30 μM PEO-IAA. Arrows indicate side branches. Scale bars: 50 μm. (**E**) Changes in branching rate following treatment of single-celled conchocelis with various concentrations of IAA in the presence of 30 μM PEO-IAA. Error bars indicate the standard deviation of triplicate independent experiments (*n* = 3), and different lowercase letters denote significant differences in the branching rate, as determined by Tukey’s test (*p* < 0.05) for each set of incubation times.

**Figure 4 cells-11-02652-f004:**
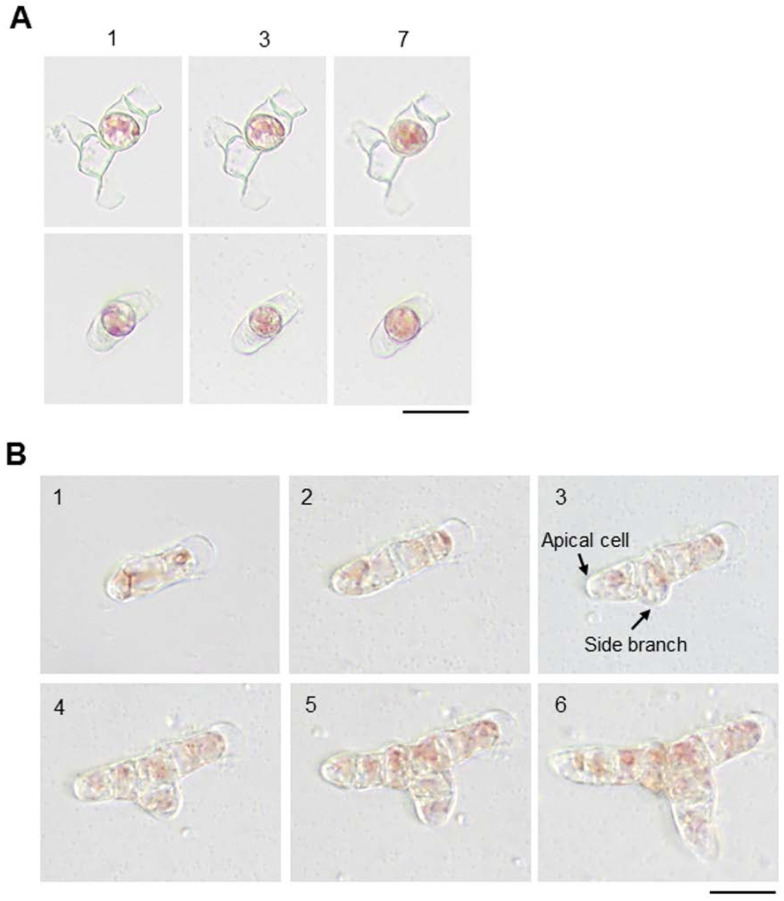
Differences in growth and side branch formation by single differentiated and apical cells from conchosporangia. (**A**) Single differentiated cells are unable to grow. Upper and lower panels show different isolated cells that failed to grow. The numbers above the panels indicate the duration of incubation (days). (**B**) Growth and production of a side branch by a single apical cell. The numbers in each panel indicate the duration of incubation (days). The apical cell and side branch after 3 days of culture are indicated by arrows. Scale bars: 25 μm.

**Figure 5 cells-11-02652-f005:**
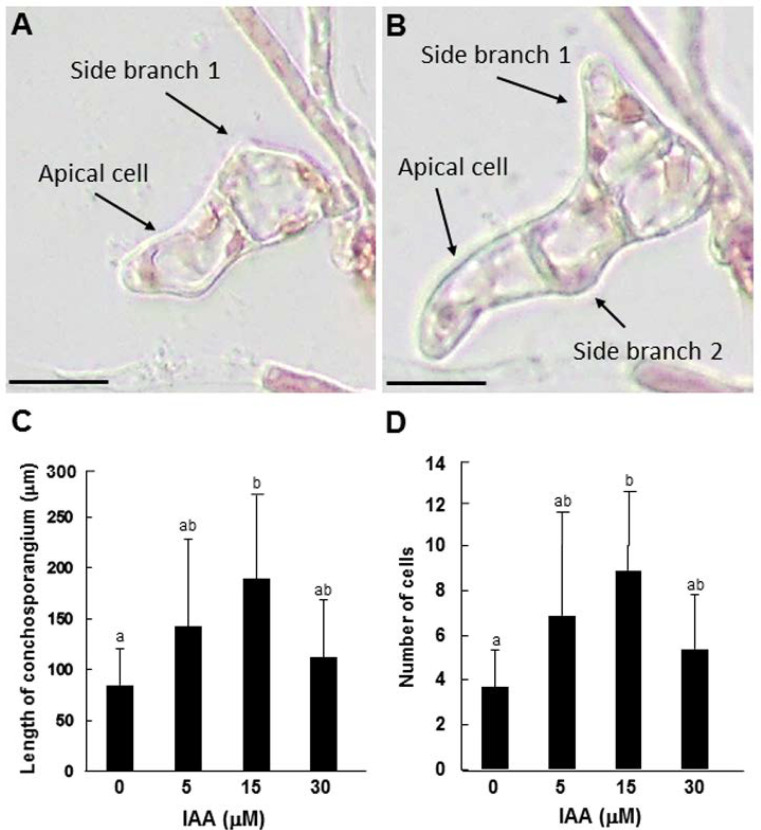
Effects of IAA on the tip growth of conchosporangia produced on conchocelis filaments. (**A**,**B**) The pattern of tip growth and side branch formation in a conchosporangium cultured for 2 (**A**) and 3 (**B**) days, after swelling of the tip of the side branch from a single-celled conchocelis. The apical cells and side branches are indicated by arrows. Scale bars: 25 μm. (**C**,**D**) Effects of exogenously supplied IAA on the growth of side branches from single-celled conchosporangia, including total length (**C**) and cell number (**D**) after 7 days of culture. Error bars indicate the standard deviation of triplicate independent experiments (*n* = 3), and different lowercase letters denote significant differences, as determined by Tukey’s test (*p* < 0.05).

**Figure 6 cells-11-02652-f006:**
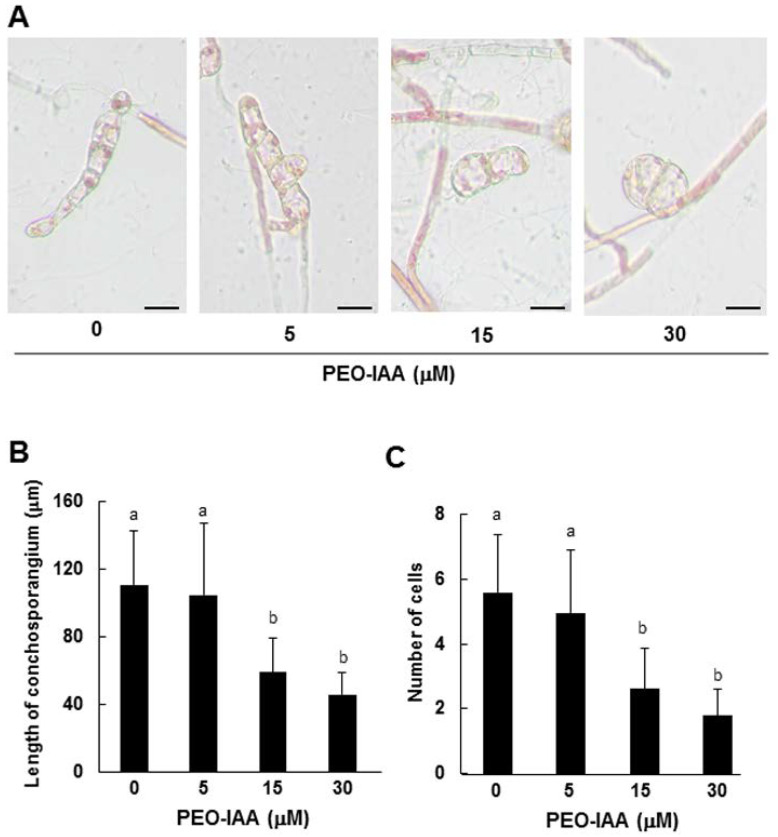
Effects of PEO-IAA on the tip growth of conchosporangia produced on conchocelis filaments. (**A**) Photographs of typical conchosporangia treated with 0, 5, 15, or 30 μM PEO-IAA for 7 days. Scale bars: 25 μm. (**B**,**C**) Effects of exogenously supplied IAA on the growth of side branches from single-celled conchosporangia, including the total length (**B**) and cell number (**C**) after 7 days of culture. Error bars indicate the standard deviation of triplicate independent experiments (*n* = 3), and different lowercase letters denote significant differences, as determined by Tukey’s test (*p* < 0.05).

**Figure 7 cells-11-02652-f007:**
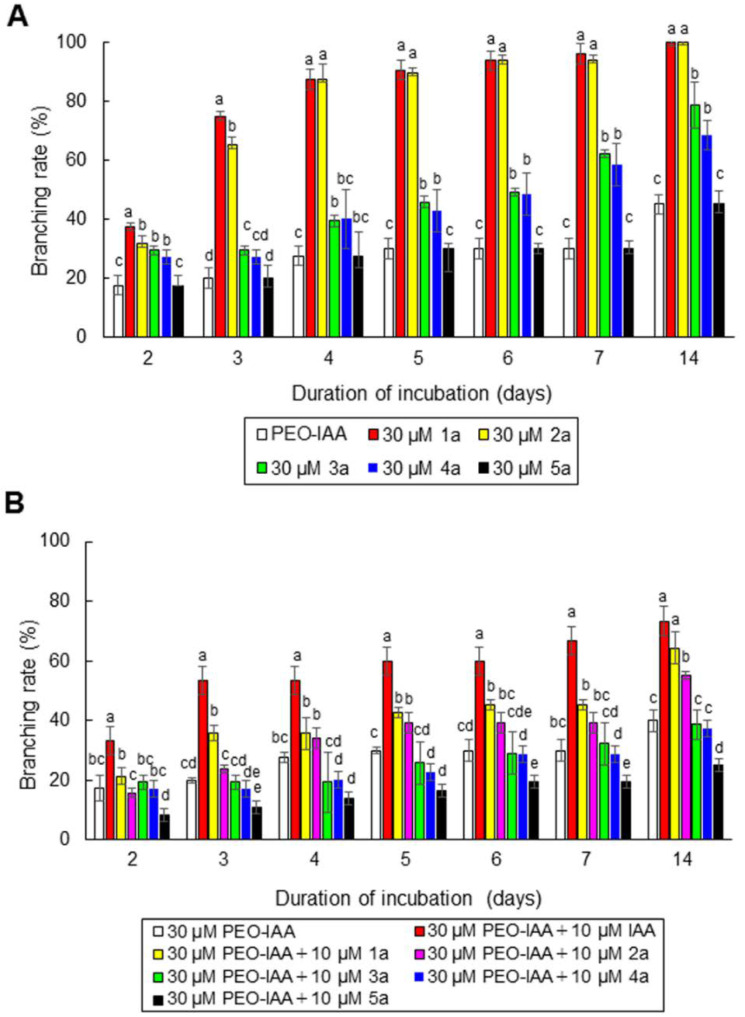
The effects of auxin and the auxin-antagonistic activity of 5-alkoxy-IAAs in *N. yezoensis*. (**A**) Comparison of the effects of 30 μM PEO-IAA and 30 μM 5-alkoxy-IAAs on the production of side branches. Side branch formation was observed in single-celled conchocelis cultured with each chemical for 2 weeks, and the number of conchocelis with branching was counted by microscopy observation at 2, 3, 4, 5, 6, 7, and 14 days to calculate the branching rate. (**B**) Activities of 5-alkoxy-IAAs. Isolated conchocelis cells were incubated in the presence of auxin derivatives **1a** to **5a** for 2 weeks, and the effects of these 5-alkoxy-IAAs on side branch formation were observed using a microscope at 1, 2, 3, 4, 5, 6, and 7 days. Error bars indicate the standard deviation of triplicate independent experiments (*n* = 3), and different lowercase letters denote significant differences in branching rate, as determined by Tukey’s test (*p* < 0.05) for each set of incubation times.

## Data Availability

Data are contained within this article or Appendix A.

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
