# Peer review of "Auxin Regulates Apical Stem Cell Regeneration and Tip Growth in the Marine Red Alga *Neopyropia yezoensis"

_cells, 2022, doi:10.3390/cells11172652_

Round 1
Reviewer 1 Report
The authors studied how auxin regulates apical stem cell regeneration and tip growth in Neopyropia yezoensis. The work addresses interesting questions in biology and will be valuable for scientists studying macroalgae and basic developmental processes in non-model organisms. The authors developed new dissection techniques to answer their questions which will be of use in future research on this topic.
The authors state that their "results provide the first evidence for the 341 physiological function of auxin in Bangiales" however it is not the first for Rhodophyta. Auxin effects were already investigated in the Rhodophyte Gracilariopsis but the authors did not mention this incredibly relevant study in their manuscript. They really need to cite and discuss the previous research performed with auxin on a Rhodophyta macroalga: "Transcriptome analysis identifies genes involved in adventitious branches formation of Gracilaria lichenoides in vitro" by Wang et al., 2005, Scientific Reports. It is a very similar study with relevant conclusions that should be compared and contrasted to the manuscript under review. Without mentioning this study or their funding sources (see below), I would not be able to recommend this interesting study for publication.
The authors state that they received no funding for this study. I cannot believe that none of the authors is receiving salary or are getting equipment and reagents for free. Please accurately state the funding for each author i.e. "Author X was supported by the xyz grant, etc."
minor points:
1) Figure S1 says "scale bars" although I could only find a single bar.
2) The authors should consider a follow-up study using single cell work. Single cell transcriptomics is a relevant technique for their research that would boost its impact and conclusions
Author Response
Thank you very much for providing your valuable comments for improvement of our manuscript. Please find our responses to your comments in the attachment file.

Reviewer 2 Report
I have reviewed the manuscript entitled “Auxin Regulates Apical Stem Cell Regeneration and Tip Growth in the Marine Red Alga Neopyropia yezoensis”. The authors investigated the critical role of auxin in tip growth in the filamentous generations of N. yezoensis. The manuscript is well written, and the authors have provided significant data to support their hypothesis. It is a topic of interest to researchers in related areas. I wish to point out only some modifications to shape the manuscript.
1. Authors should revise the clear objectives.
2. The Genus and Species name should be italic in the entire manuscript.
3. The authors should improve the quality of the figures.
4. The authors should add more discussion with recent references.
5. Conclusion: Please provide some major data. The conclusion should be supported by data.
6. This manuscript has some types of misspellings. It is imperative that these are corrected and the language should be improved.
Author Response

(The authors gave the same response as above.)
